# Single-Cell RNA-Seq Reveals Transcriptomic Heterogeneity and Post-Traumatic Osteoarthritis-Associated Early Molecular Changes in Mouse Articular Chondrocytes

**DOI:** 10.3390/cells10061462

**Published:** 2021-06-10

**Authors:** Aimy Sebastian, Jillian L. McCool, Nicholas R. Hum, Deepa K. Murugesh, Stephen P. Wilson, Blaine A. Christiansen, Gabriela G. Loots

**Affiliations:** 1Lawrence Livermore National Laboratory, Physical and Life Science Directorate, Livermore, CA 94550, USA; mccool1@llnl.gov (J.L.M.); hum3@llnl.gov (N.R.H.); murugesh2@llnl.gov (D.K.M.); wilson265@llnl.gov (S.P.W.); 2School of Natural Sciences, University of California Merced, Merced, CA 95343, USA; 3Department of Orthopaedic Surgery, University of California Davis Health, Sacramento, CA 95817, USA; bchristiansen@ucdavis.edu

**Keywords:** osteoarthritis, chondrocyte heterogeneity, scRNA-seq, PTOA, cartilage, gene expression, knee injury

## Abstract

Articular cartilage is a connective tissue lining the surfaces of synovial joints. When the cartilage severely wears down, it leads to osteoarthritis (OA), a debilitating disease that affects millions of people globally. The articular cartilage is composed of a dense extracellular matrix (ECM) with a sparse distribution of chondrocytes with varying morphology and potentially different functions. Elucidating the molecular and functional profiles of various chondrocyte subtypes and understanding the interplay between these chondrocyte subtypes and other cell types in the joint will greatly expand our understanding of joint biology and OA pathology. Although recent advances in high-throughput OMICS technologies have enabled molecular-level characterization of tissues and organs at an unprecedented resolution, thorough molecular profiling of articular chondrocytes has not yet been undertaken, which may be in part due to the technical difficulties in isolating chondrocytes from dense cartilage ECM. In this study, we profiled articular cartilage from healthy and injured mouse knee joints at a single-cell resolution and identified nine chondrocyte subtypes with distinct molecular profiles and injury-induced early molecular changes in these chondrocytes. We also compared mouse chondrocyte subpopulations to human chondrocytes and evaluated the extent of molecular similarity between mice and humans. This work expands our view of chondrocyte heterogeneity and rapid molecular changes in chondrocyte populations in response to joint trauma and highlights potential mechanisms that trigger cartilage degeneration.

## 1. Introduction

Osteoarthritis (OA) is a degenerative joint disorder that affects more than 300 million people worldwide, often resulting in diminished quality of life and disability [1,2]. Although OA prevalence is on the rise and the Center for Disease Control estimates that as many as 78 million Americans (or one in four) will suffer from OA by 2040, an in-depth understanding of the joint microarchitecture and molecular mechanisms that contribute to OA initiation and progression is still in its infancy [1]. Furthermore, the lack of sufficient progress in this area has severely hindered the development of effective therapeutic approaches for the early diagnosis, prevention, and treatment of OA.

The knee joint is a complex structure composed of several tissues including articular cartilage, synovial membrane, joint capsule, menisci, subchondral bone, infrapatellar and suprapatellar fat pads, and tensile connective tissues including tendons and ligaments [3]. Degeneration of the articular cartilage, the connective tissue lining the articular surfaces of synovial joints, is a hallmark of OA. This type of cartilage has unique viscoelastic properties with an extraordinary capacity to withstand high cyclic loads, but it is also avascular with limited potential to self-renew [4]. It is composed of a densely packed extracellular matrix (ECM) with a sparse distribution of chondrocytes of varying morphology and number depending on the anatomical position within the articular cartilage [5]. While distinct chondrocyte morphologies have been recognized histologically, we have yet to determine the molecular profiles underlying chondrocyte heterogeneity to further our understanding of how specific transcripts corroborate chondrogenic phenotypes in the healthy joint as well as how perturbations in individual cells mediate OA pathology.

Several studies have transcriptionally profiled articular cartilage/chondrocytes using human tissue samples obtained during joint replacement surgeries, but generally these results represent late-stage OA cartilage only [6,7,8,9], and may lack details on disease etiology. Although these studies yielded new insights into the molecular programs involved in OA progression, they provided limited information about the molecular profiles of chondrocytes isolated from healthy joints. These studies were also unable to distinguish early from late molecular changes associated with cartilage degeneration. Animal models are widely used to study joint biology and offer a unique opportunity to study early stages of OA and mechanisms of disease progression. In recent years, several groups, including ours, have employed techniques such as bulk tissue RNA sequencing (RNA-seq) and microarrays to investigate the molecular changes in the mouse joint that contribute to the development of OA [10,11,12,13]. However, whole-joint gene expression profiling does not provide any information about the underlying cellular heterogeneity and fails to distinguish cell type-specific changes versus transcriptional changes happening in multiple cell types. Although several studies have attempted to profile specific components of micro-dissected joints to obtain more ‘tissue-specific’ gene expression profiles, such undertakings have been technically challenging and likely generate samples contaminated by some of the adjacent tissues [14,15]. Single-cell RNA sequencing (scRNA-seq) enables us to overcome these limitations [16] by allowing us to profile gene expression in all individual cells purified from a complex structure such as the knee joint. This will in turn help us understand molecular differences between various cell types and subtypes as a function of disease progression and will lead to the discovery of previously unknown cell populations, cell type-specific transcriptional profiles, and molecular pathways responsible for the development of OA.

Isolating chondrocytes from the articular cartilage is technically challenging due to the cells being embedded within a dense, fibrous ECM [17]. In the present study, we successfully isolated articular chondrocytes from adult mouse knee joints before and after traumatic injury, and profiled their transcriptomes using scRNA-seq. Through this approach we identified nine transcriptionally distinct chondrocyte subpopulations and determined the injury-induced molecular changes in these subpopulations within 7 days post-injury. Furthermore, we compared the repertoire of mouse chondrocyte subtypes to human chondrocytes harvested from OA cartilage to evaluate the extent of cross-species translatability of pre-clinical results from animal models of OA. We identified six mouse chondrocyte subtypes with highly molecular fidelity to the human subtypes. This study allowed us to elucidate the full extent of heterogeneity among articular chondrocytes in healthy and injured mouse cartilage, and identify several chondrocyte subtypes with defined markers. The injury-induced early molecular changes described in these chondrocytes may represent new therapeutic targets for evaluating therapeutic interventions for the prevention of cartilage degeneration in response to traumatic injury.

## 2. Materials and Methods

### 2.1. Anterior Cruciate Ligament (ACL) Injury Model

Ten-week-old male C57Bl/6J (BL6) mice (purchased from Jackson Laboratory Bar Harbor, ME, USA; Stock No: 000664) were subjected to anterior cruciate ligament injury using a single non-invasive tibial compressive overload, as previously described [12,13,18,19]. Tibial compression was applied at 1 mm/s, and loading was manually stopped immediately after ACL injury.

### 2.2. Histological Assessment of Disease Severity

Knee joints were collected from uninjured, 3 days post injury (DPI), 7DPI, and 6-weeks post injury (*n* ≥ 3/group) mice and processed for histological evaluation as previously described [12]. Briefly, whole joints were fixed in 10% neutral buffered formalin (NBF), decalcified using 0.5 M ethylenediamine tetraacetic acid (EDTA), and processed for paraffin embedding. Joints were sectioned in the sagittal plane at 6 µm and serial medial sections that included the femoral condyles, menisci, and tibial plateaus were prepared for histological assessment of joint tissue integrity. Sections were stained on glass slides using 0.1% Safranin-O (0.1%, Sigma, St. Louis, MO, USA; S8884) and 0.05% Fast Green (0.05%, Sigma, St. Louis, MO, USA; F7252) using standard procedures (IHC World, Woodstock, MD, USA), and then imaged using a Leica DM5000 microscope.

### 2.3. Immunohistochemistry (IHC)

Sagittal sections from uninjured, 3DPI, and 7DPI knee joints of BL6 mice were used for IHC (n ≥ 3/group). Primary antibodies were incubated overnight at 4 °C in a dark, humid chamber following antigen retrieval. Secondary antibodies were incubated for 2 h at room temperature in a dark, humid chamber at 1:500. Negative control slides were incubated with secondary antibody only. Stained slides were mounted with Prolong Gold with DAPI (Molecular Probes, Eugene, OR, USA). Slides were imaged using a Leica DM5000 microscope (Leica Microsystems, Wetzlar, Germany). ImagePro Plus V7.0 Software, a QIClick CCD camera (QImaging, Surrey, BC, Canada), and ImageJ V1.53 Software were used for imaging and photo editing. Primary antibodies included: CYTL1 (Proteintech, Rosemont, IL, USA; 15856-1-AP (1:75)); MATN3 (R&D, Minneapolis, MN, USA; AF3357 (1:100)); SPP1 (Abcam, Cambridge, UK; ab218237 (1:100)); MMP3 (Abcam, Cambridge, UK; ab52915 (1:100)); CHIL1 (Thermofisher, Waltham, MA, USA; MA5-36122 (1:100)); and INHBA (Thermofisher, Waltham, MA, USA; 10651-1-AP (1:100)). Secondary antibodies included: Chicken anti-rabbit 488 (Thermofisher, Waltham, MA, USA; A21441), Chicken anti-rabbit 594 (Thermofisher, Waltham, MA, USA; 21442), and Donkey anti-goat 594 (Thermofisher, Waltham, MA, USA; A11058).

### 2.4. Single-Cell RNA Sequencing (scRNA-seq)

Uninjured, 3DPI, and 7DPI joints (*n* = 5/group) were used for scRNA-seq analysis. Mice were euthanized and hindlimbs were collected by removing the legs at the hip joint and storing on ice in Dulbecco’s Modified Eagle Medium Nutrient Mixture F-12 (DMEM/F-12) (Thermo Fisher Scientific, Waltham, MA, USA). Articular cartilage from tibia and femora was isolated by cutting ~1 mm of tissue from the end of both long bones at the knee joint. For each experimental group, cartilage tissue from 5 mice was pooled, and digested to a single-cell suspension homogenate in 5 mL of 0.2% Collagenase 2 solution (2 mg/mL Thermo Fisher Scientific, Waltham, MA, USA) while shaking at 37 °C for a total of 2 h in 30-minute intervals. After each 30-minute interval, fractions were filtered through a 70 μm Nylon cell strainer into DMEM/F12 with 10% fetal bovine serum (FBS) and kept on ice. Remaining undigested cartilage tissue was further digested in 5 mL of fresh Collagenase 2 digestion media. After the final digestion interval, cells were pelleted via centrifugation for 10 min at 500 G at 4 °C, and incubated on ice with ACK lysis buffer (Thermo Fisher Scientific, Waltham, MA, USA) to remove red blood cells. Cells were stained with the following antibodies for flow cytometry and fluorescently activated cell sorting (FACS) analysis: CD45 APC-Cy7 (BioLegend, San Diego, CA, USA, 103116 (1:100)), Ter119 APC (Miltenyi Biotec, Bergisch Gladbach, Germany, 130-102-290, (1:10)), and DAPI (Thermo Fisher Scientific, Waltham, MA, USA). Immune and erythroid contamination was depleted via double-negative selection (CD45^−^; Ter119^−^) using the BD FACSMelody (San Jose, CA, USA). Final cell counts after FACS were performed manually using a hemocytometer and then resuspended in PBS + 0.04% nonacetylated BSA for preparation of scRNA-seq using a Chromium Controller (10× Genomics, Pleasanton, CA, USA). Library preparation was performed using Chromium Single Cell 3′ GEM, Library & Gel Bead Kit v3 (10× Genomics, Pleasanton, CA, USA; Catalog no. 1000075) following the manufacturer’s protocol and sequenced using Illumina NextSeq 500 targeting approximately 50,000 reads per cells.

### 2.5. scRNA-seq Data Analysis of Chondrocytes from Uninjured Joints

Alignment of scRNA-seq data to the mouse genome (mm10) and gene counting was completed utilizing the 10× Genomics Cell Ranger pipeline (10× Genomics, Pleasanton, CA, USA). Subsequently, output files from the Cell Ranger ‘count’ were read into Seurat v3 [20] for further analysis. Cells with fewer than 500 detected genes or genes that were expressed by fewer than 5 cells were excluded from the analysis. Dead cells and doublets were also removed as previously described [21]. After removing all the unwanted cells from the dataset, the data was normalized by employing a global-scaling normalization method ‘LogNormalize’. Subsequently, the 2000 most variable genes were identified, the data were scaled, and the dimensionality of the data was reduced by principal component analysis (PCA). Subsequently, we constructed a K-nearest-neighbor (KNN) graph based on the Euclidean distance in PCA space using the ‘FindNeighbors’ function and applied Louvain algorithm to iteratively group cells together by the ‘FindClusters’ function. A non-linear dimensional reduction was then performed via uniform manifold approximation and projection (UMAP) and various cell clusters were identified. Then, clusters expressing immune and blood cell markers were removed and the remaining data were normalized, scaled, and, after variable feature identification, the data were re-clustered to identify clusters of non-immune cells in the joint. To identify chondrocyte subtypes, clusters expressing chondrocyte markers *Acan*, *Sox9*, and *Col2a* were extracted and further analyzed as described above. Marker genes per cluster were calculated using Seurat’s ‘FindAllMarkers’ function and the ‘wilcox’ test option. All scRNA-seq data described herein were deposited in the Gene Expression Omnibus (GEO) database, GEO accession ID GSE172500.

### 2.6. Analysis of Human Chondrocyte scRNA-seq Data

Human chondrocyte scRNA-seq data [6] were downloaded from Gene Expression Omnibus (GEO) database (GSE104782) and a text file was obtained with raw expression values. The data were analyzed using Seurat [6], as described above to identify various cell types. Subclusters were annotated based on the markers provided by Ji et al. [6].

### 2.7. Comparison of Chondrocytes from Uninjured and Injured Joints

scRNA-seq data from uninjured, 3DPI, and 7DPI joints were analyzed using Seurat v3 [6]. After data pre-processing, variable features were selected based on a variance stabilizing transformation (‘vst’). Then, we identified anchors for data integration using the ‘FindIntegrationAnchors’ function. Next, these anchors were passed to the ‘IntegrateData’ function and new integrated matrix with all 3 datasets were generated. Subsequent dimensionality reduction, clustering, and visualization were performed in Seurat as described above. Clusters of cells expressing the chondrocyte markers *Sox9*, *Acan*, and *Col2a1* were extracted and further analyzed to identify various chondrocyte subpopulations. Genes differentially expressed between chondrocyte subtypes at various timepoints were identified using ‘FindMarkers’ function implemented in Seurat.

### 2.8. Pseudotime Trajectory Finding

Pseudotime trajectory of chondrocytes was constructed with Monocle [22]. Expression data, phenotype data, and feature data were extracted from the Seurat object and a Monocle ‘CellDataSet’ object was constructed. Highly variable genes from Seurat object were used as ordering genes in Monocle. Dimensionality reduction was performed using the DDRTree algorithm implemented Monocle via the ‘reduceDimension’ function. Cells were ordered along the trajectory using the ‘orderCells’ method with default parameters.

### 2.9. Ontology Enrichment Analysis

Genes enriched in each chondrocyte subtype were identified using ‘FindAllMarkers’ function from Seurat [6]. Genes with >1.25-fold enrichment in each cluster with an adjusted *p*-value < 0.1 were used for ontology enrichment analysis using ToppGene Suite [23] and Metascape [24]. Dot plots of enriched ontology terms were generated in R using the ggplot2 package.

## 3. Results

### 3.1. Single-Cell Profiling Reveals Cellular Heterogeneity in Healthy Murine Knee Joints

To characterize articular chondrocyte heterogeneity, we profiled uninjured knee cartilage from adult mouse joints at a single-cell resolution. About 1 mm wide articular cartilage tissue was dissected from the ends of tibia and femora, enzymatically digested to a single cell suspension, and depleted of immune and blood cells to obtain a chondrocyte-enriched (CD45^−^; Ter119^−^) cell fraction that was subjected to sequencing (Figure 1A). Unsupervised clustering of the data resulted in 13 cell type clusters including some remnant immune (CD45^+^/Ptprc^high^) and blood cell (Hemgn^high^) clusters (Appendix A). These immune and blood cells were computationally filtered and excluded from subsequent analysis. Analysis of the remaining 2490 cells resulted in 10 cell clusters with distinct gene expression profiles (Figure 1B and Appendix A). Cell type identities were assigned based on previously published cell-type specific markers [21,25,26,27,28,29,30,31,32,33] (Figure 1B). Cells in clusters 0, 1, 3, and 4 expressed high levels of chondrocyte markers *Sox9*, *Col2a1*, and *Acan* [25] and were labeled ‘chondrocytes’ (Figure 1C,D). Cluster 7 also expressed high levels of *Sox9*, *Col2a1*, and *Acan.* However, this cluster also showed enrichment for several cell cycle genes, including *Mki67*, *Cdk1*, *Stmn1*, *Top2a*, and *Cenpa* [26]; this cluster was therefore annotated as ‘proliferating chondrocytes’ (Figure 1D and Appendix A). Cluster 2 expressed markers of synovial subintimal fibroblasts (SSF) including *Cxcl12*, *Col3a1*, and *Col14a1* [7] along with many other mesenchymal and fibroblast markers including *Pdgfra*, *Pdpn*, *Clec3b*, *Abi3bp*, *Col3a1*, *Col14a1*, fibroblast-specific protein 1 (*FSP1*/*S100a4*), and *Thy1*, and was labeled as the ‘mesenchymal cells/fibroblasts’ cluster (Figure 1D and Appendix A) [21,27,28]. As we previously reported for fibroblasts isolated from the mammary fat pad [21], fibroblast/mesenchymal cells from the joint also expressed cytokines *Ccl2*, *Ccl7*, *Cxcl1*, and *Cxcl12* and complement pathway genes *C3* and *C4b* (Appendix A). Cluster 5 was labeled ‘osteoblasts’ based on enrichment of the well-known osteoblast markers *Col1a1*, *Col1a2*, osteocalcin (*Bglap*), and alkaline phosphatase (*Alpl*) (Figure 1D) [29]. Cluster 6 showed high expression of endothelial cell markers *Pecam1*, *Ptprb*, and VE-Cadherin (*Cdh5*) [21,30] whereas cluster 8 showed enrichment for pericyte markers including *Rgs5*, *Myh11*, and *Mcam* [28,31,32,33], and were classified as ‘endothelial’ and ‘pericytes’, respectively (Figure 1D). We also identified a small cluster of ‘synovial intimal fibroblasts’ (SIFs). SIFs expressed fibroblast markers *Pdgfra* and *Pdpn*, along with high levels of the SIF markers lubricin (*Prg4*), *Has1*, and *Htra1* (Figure 1D) [7]. A cluster tree revealed the relationship between these cell populations. As expected, all chondrocyte clusters showed tight similarities in their expression profiles (Appendix A). Osteoblasts and mesenchymal cells/fibroblasts were found closer to each other, while endothelial cells and pericytes appeared to be more similar to each other (Appendix A), than to all other cluster types. SIFs were distinct from all other cells and formed a separate branch on the cluster tree (Appendix A).

Next, we compared the gene expression profiles of chondrocytes to all other cell type clusters (clusters 2, 5, 6, 8, 9) to identify genes enriched in healthy chondrocytes compared to other connective tissue-forming cells in the joint (Appendix A, Appendix A). The master chondrocyte transcription factor *Sox9* [25] and several other transcription factors (*Nfatc2*, *Runx3*, *Bhlhe40*, *Bhlhe41*, *Lef1*, *Tsc22d1*, and *Sox5*) showed significant enrichment in all chondrocyte clusters, suggesting that these transcription factors play major roles in regulating chondrocyte differentiation or homeostasis (Figure 1E and Appendix A). Previously, it has been shown that the ECM of articular cartilage is distinct from other types of connective tissues [4]. In addition to the well-established chondrocyte marker *Col2a1*, we identified several collagens differentially expressed by chondrocytes relative to other cell types including *Col9a1*, *Col9a2*, *Col9a3*, *Col11a1*, *Col11a2*, *Col12a1*, and *Col27a1* (Appendix A). Chondrocytes also showed enrichment for numerous proteoglycans (*Acan*, *Chad*, *Epyc*, *Prelp*, and *Hapln1*) and glycoproteins (*Cilp*, *Cilp2*, *Crispld1*, *Thbs1*, *Matn3*, *Comp*, and *Smoc2*) (Appendix A). Several members of the serpin family (*Serpina1a*, *Serpina1b*, *Serpina1d*, and *Serpine1*), *Smpd3* (an enzyme involved in sphingolipid metabolism) [34], and *Papss2* and *Chst11* (enzymes involved in the glycosaminoglycan metabolic process) [35,36] also showed significantly higher expression in chondrocytes compared to other cell types from the murine articular joint (Appendix A). This analysis provides novel insights into cellular heterogeneity in synovial joints and highlights differences in the transcriptome of articular chondrocytes relative to other connective tissues in the healthy knee.

### 3.2. Identification of Potential OA Targets Enriched in Chondrocytes

OA is a complex disease which affects multiple tissue types in the joint. Several human and animal model-based bulk gene expression studies have identified thousands of genes dysregulated in OA joints. However, for many of these genes the tissue/cell types that express these genes are not known. To identify OA-associated genes that are primarily expressed by chondrocytes we obtained a list of potential OA targets from OAtarget, a recently developed knowledgebase of genes associated with OA, and determined their expression levels in various cell types in the mouse joint [37]. A total of 62 OA target genes identified by 3 or more independent studies were ≥2-fold enriched in chondrocytes compared to other cell types (Figure 2A, Appendix A). SIFs showed ≥2-fold enrichment for 60 genes, while mesenchymal cells/fibroblasts showed enrichment for 29 OA targets (Appendix A). Twenty-six OA targets were enriched in osteoblasts. We also noted that both pericytes and endothelial cells showed enrichment for ~100 potential OA targets (Appendix A), suggesting the complex molecular contribution of many different cell types to OA pathology.

Although many of the potential OA targets were expressed by multiple cell types at a comparable level, significant enrichment for a select group of these genes in chondrocytes could indicate their direct involvement in regulating cartilage metabolism and other cellular processes. Potential OA targets significantly enriched in chondrocytes compared to other cell types included *Col2a1*, *Acan*, *Cnmd*, *Chad*, *Scrg1*, *Wwp2*, cartilage intermediate layer protein (*Cilp*), *Smpd3*, osteoprotegerin (OPG/*Tnfrsf11b*), Wnt inhibitory factor 1 (*Wif1*), Matrilin 3 (*Matn3*), *Col9a1-3*, and *Hapln1* (Figure 2A,B). We also observed that some of these genes had their expression restricted to specific chondrocyte subpopulations, which points to a substantial heterogeneity within chondrocyte subtypes. Among chondrocytes, cluster 0 showed enrichment for *Matn3*, *Col9a1-3*, and *Hapln1*, whereas cluster 1 showed enrichment for cytokine-like 1 (*Cytl1*), *Bmp2*, *Ibsp*, and cartilage intermediate layer protein 2 (*Cilp2*) (Figure 2A,C, Appendix A). This cluster also showed significant enrichment for OA targets including osteoprotegerin (OPG/*Tnfrsf11b*), *Wif1*, and *Cilp* (Figure 2B). Cluster 3 showed enrichment for *Krt16*, *M1ap*, and *Dusp5*, whereas cluster 4 showed enrichment for chitinase-like 1 (*Chil1*), *Mgp*, and *Apoe* (Figure 2C and Appendix A). To define chondrocyte subtypes more in depth, we next analyzed the transcriptional profiles of chondrocyte clusters separately.

### 3.3. scRNA-seq Analysis Identified Nine Chondrocyte Subtypes in Mouse Knee Joints

All cells from clusters expressing chondrocyte markers (clusters 0, 1, 3, 4, and 7) were extracted and re-examined in greater detail. This included 1625 cells expressing high levels of *Sox9*, *Col2a1*, and *Acan* (Figure 3A), accounting for ~65% of all stromal cells analyzed. The analysis of these chondrocytes revealed nine clusters with distinct gene expression profiles (Figure 3B–D and Appendix A, Appendix A). Cluster 0 showed enrichment for *Ucma*, *Matn3*, *Papss2*, and *Scrg1* (Figure 3C,D and Appendix A, Appendix A) and was annotated ‘Ucma^high^’. These genes were also enriched in clusters 2, 3, and 6, but these clusters had additional cluster-specific markers (Appendix A). Cluster 2 showed enrichment for *Mef2c*, *Ihh*, and *Pth1r*, markers of pre-hypertrophic chondrocytes [38,39,40], and this cluster was named ‘Mef2c^high^’ (Figure 3C,D and Appendix A). A subset of cell from the Mef2c^high^ cluster also expressed *Col10a1*, a marker of chondrocyte hypertrophy [38] (Appendix A). Cluster 3 had unique markers *Krt16*, *M1ap*, *Ngf*, and *Srxn1* and was annotated as ‘Krt16^high^’ (Figure 3C,D and Appendix A, Appendix A). Cluster 6 expressed high levels of cell cycle-associated genes *Cdk1*, *Top2a*, *Cenpf*, and *H2afz*, and was identified as ‘dividing chondrocytes’ (divC).

Cluster 1 expressed high levels of cytokine-like 1 (*Cytl1*), *Cilp2*, and *Prg4*, and was annotated ‘Cytl1^high^’ (Figure 3C,D and Appendix A). Clusters 7 and 8 also robustly expressed these genes, but cluster 7 showed enrichment for additional genes including *Tnfaip6*, *Smoc2*, *Clu*, and *Gas1*, whereas cluster 8 showed enrichment for fibroblast/fibrosis markers, fibroblast-specific protein 1 (*S100a4/FSP1*), and *Col1a1*, *Col3a1*, and *Abi3bp*, in addition to *Npy* and osteopontin (OPN)/secreted phosphoprotein 1 (*Spp1*) (Figure 3C,D and Appendix A). Cluster 7 was annotated as ‘Tnfaip6^high^’ and cluster 8 was annotated as ‘S100a4^high^’. Tnfaip6^high^ cluster also expressed many fibroblast markers and regulators of fibrosis including *Abi3bp*, *Inhba*, and *Spp1*, but the expression levels for many of these genes were not as high as in the S100a4^high^ cluster [41,42] (Appendix A and Appendix A). It is likely that the Tnfaip6^high^ cluster represents pre-fibrotic chondrocytes and the S100a4^high^ cluster represents chondrocytes with a more mature fibrotic phenotype (Appendix A). Cluster 4 expressed high levels of *Chil1* (CHI3L1), Efemp1, *Spon1*, *Mgp*, and *Fxyd3*, and was annotated ‘Chil1^high^’ (Figure 3C,D and Appendix A). Cluster 5 shared several markers with other subtypes but also had higher expression of *Neat1*, *Malat1*, *Ogt*, and *Wwp2*, and was denoted the ‘Neat1^high’^ cluster (Appendix A, Appendix A).

To understand the relationship among these chondrocyte subpopulations, we constructed a transcriptional trajectory of these cells on a pseudotime scale using Monocle [22]. Pseudotemporal trajectory analysis predicted a branched trajectory where divCs resided at one end, while Tnfaip6^high^ and S100a4^high^ cells were observed at the opposing end of the trajectory (Figure 3E,F). Cytl1^high^ cells were closer to the Tnfaip6^high^ and S100a4^high^ clusters. Cells from the Chil1^high^ and Neat1^high^ clusters resided along the trajectory (Figure 3D,E), while the Krt16^high^, and Mef2c^high^ clusters formed distinct branches (Figure 3E). Ucma^high^ cells resided closer to cells from the Krt16^high^ and Mef2c^high^ clusters and DivCs. This analysis suggested that the Tnfaip6^high^, S100a4^high^, and Cytl1^high^ clusters were closer in developmental time while Krt16^high^ and Mef2c^high^ clusters were developmentally closer to the Ucma^high^ cluster, which begged the question as to whether the physical relationship to each other translated to similarity in localization or function.

### 3.4. Molecular and Functional-Level Characterization of Chondrocyte Subpopulations

To further understand the potential functional roles of these chondrocyte subtypes in maintaining articular cartilage metabolism and joint homeostasis, we performed a gene ontology enrichment analysis with genes upregulated in each chondrocyte subpopulation relative to all other subtypes (Appendix A). As expected, all chondrocyte subtypes showed significant enrichment for the ontology terms ‘cartilage development’ and ‘ECM organization’ (Figure 4A). However, many of the cartilage ECM genes were differentially expressed among the subtypes. For example, collagens *Col9a1*-*a3*, *Col11a1*-*a2*, and *Col12a2* showed significant enrichment in the Ucma^high^, Krt16^high^, and Mef2c^high^ clusters, whereas *Col1a1*, *Col1a2*, and *Col3a1* showed enrichment in the Cytl1^high^ and S100a4^high^ clusters (Appendix A). In addition to the Mef2c^high^ cluster, *Col10a1* expression was detected in the Tnfaip6^high^ cluster (Appendix A). Collagen-processing enzymes such as *Lox*, *Loxl1*, and *P3h2* were also differentially expressed between these chondrocyte subtypes (Appendix A). We also identified several proteoglycans and proteoglycan-processing enzymes differentially expressed among these chondrocyte subtypes (Appendix A). Notably, *Acan*, *Bgn*, *Epyc*, *Hapln*, *Chadl*, *Gpc1*, and two enzymes involved in proteoglycan metabolism, *Papss2* and *Chst12*, were significantly enriched in the Ucma^high^, Krt16^high^, and Mef2c^high^ clusters (Figure 3A and Appendix A). Proteoglycans lubricin (*Prg4*), *Dcn*, and *Chad* showed enrichment in the Cytl1^high^, Tnfaip6^high^, and S100a4^high^ clusters, and *Lum*, *Srgn*, *Ogn*, and *Omd* showed enrichment in the Chil1^high^ cluster (Appendix A). We also found that genes enriched in Chil1^high^ cluster had a minimal overlap with other clusters, suggesting that this cluster has a transcriptome profile distinct from all other articular chondrocytes (Figure 4B).

The Ucma^high^ and Krt16^high^ clusters also showed enrichment for ontology terms ‘mRNA metabolism’ and ‘translation’ and had high expression of several ribosomal proteins (Figure 4A, Appendix A). Biological processes such as ‘cell adhesion’ and ‘cell migration’ were enriched in all clusters except Ucma^high^ and Mef2c^high^ (Figure 4A). We also found that the Mef2c^high^ cluster showed enrichment for many genes associated with ‘biomineralization’ including *Alpl*, *Sp7*, *Bglap*, and *Pth1r*, in addition to Mef2c (Figure 4A and Appendix A). Some of these genes are also associated with pre-hypertrophic/hypertrophic chondrocyte phenotypes and may also reside in the growth plate [38,39,40,43].

The Wnt signaling pathway was enriched in the Chil1^high^ cluster (Figure 4A). This cluster showed enrichment for Wnt pathway inhibitors *Notum*, *Sfrp5*, *Dact1*, and Wnt inhibitory factor 1 (*Wif1*) (Appendix A, Appendix A). Several other members of the Wnt signaling pathway including *Lef1*, *Daam2*, and *Fzd9* were also enriched in this cluster (Appendix A, Appendix A). Interestingly, the Mef2c^high^ cluster showed enrichment for *Wnt5b*, but Wnt4 had the highest expression in cells from the Krt16^high^ cluster (Appendix A, Appendix A). The Chil1^high^ cluster also showed enrichment for cytokines *Cxcl14*, *Il17d*, and *Apoe* (Appendix A, Appendix A), but Il17b was robustly expressed by Mef2c^high^, Ucma^high^, Krt16^high^, and divCs chondrocytes (Appendix A). Additionally, the Krt16^high^ cluster showed enrichment for *Il11* (Appendix A, Appendix A). These results suggest that active signal transduction networks may be initiated between individual chondrocyte subpopulations and they may be responsible for specific biological functions in joint crosstalk to bone and other tissues.

Several genes associated with ‘mesenchymal cell proliferation’ including *Prrx1*, *Bmp2*, and *Nifb* were enriched in the Cytl1^high^, Tnfaip6^high^, and S100a4^high^ clusters (Figure 4A and Appendix A). These clusters also showed enrichment for TGF-beta/BMP signaling pathway genes including *Bmp2*, *Tgfbr1*, *Acvr1*, and *Inhba* (Appendix A). In addition, the Tnfaip6^high^ cluster showed enrichment for genes associated with Foxo signaling including *Foxo1*, *Sirt1*, and *Irs1* (Appendix A). Foxo1 was also enriched in the S100a4^high^ cluster (Appendix A). Additionally, the S100a4^high^ cluster showed enrichment for several fibroblast/fibrosis-associated genes including *S100a4*, *Col3a1*, *Ly6c1*, *Abi3bp*, and *Dcn*, along with genes such as *Sod3*, an enzyme involved in reactive oxygen species degradation and *Mustn1*, a regulator of tissue regeneration (Appendix A, Appendix A). The Neat1^high^ cluster showed enrichment for genes involved in RNA processing and splicing such as *Fus*, *Luc7l2*, *Malat1* and *Tra2a* (Appendix A; Appendix A). It also showed enrichment for *Wwp2*, an HECT-type E3 ubiquitin ligase that plays a major role in maintenance of cartilage homeostasis (Appendix A) [44].

Next, we performed immunohistochemistry (IHC) with antibodies targeting four chondrocyte subtype markers marking distinct populations. Cytl1, which was robustly expressed in Cytl1^high^ and Tnfaip6^high^ clusters (Figure 3D and Figure 4C) was primarily expressed in the mid layer of the articular cartilage with minimal to no expression in the superficial layer, whereas Matn3, which was robustly expressed in Ucma^high^, Mef2c^high^, Krt16^high^ clusters, and divCs (Figure 4C and Appendix A), showed significant protein expression in the chondrocytes residing in the superficial layer (Figure 4D). Matn3 and Cytl1 expression was also detected in few cells from deep/calcified layers. Chil1, which was highly enriched in the Chil1^high^ cluster (Figure 3D and Figure 4C), was primarily detected in a subset of superficial and mid-layer chondrocytes (Figure 4E). Spp1, which was highly enriched in S100a4^high^ cluster (Figure 4C and Appendix A), was expressed by chondrocytes in the deep/calcified layers of the articular cartilage (Figure 4F). Robust Spp1 expression was also detected in the bone (Figure 4F). These findings suggest that chondrocytes from different regions of the articular cartilage have distinct molecular profiles and potentially different functions.

### 3.5. Comparative Transcriptomic Analysis Identified Similarities and Differences between Mouse and Human Chondrocyte Subtypes

Ji et al. recently profiled human osteoarthritic chondrocytes using scRNA-seq and identified seven major chondrocyte subtypes labeled proliferative chondrocytes (proCs), pre-hypertrophic chondrocytes (pre-HTCs), hypertrophic chondrocytes (HTCs), fibrocartilage chondrocytes (FCs), effector chondrocytes (ECs), regulatory chondrocytes (RegCs), and homeostatic chondrocytes (HomCs) [6]. Here, we compared the mouse chondrocyte scRNA-seq data to this human chondrocyte scRNA-seq dataset. Re-analysis of the human OA chondrocytes resulted in seven chondrocyte subclusters (Figure 5A,B).

*Cytl1* was described as a gene highly expressed in ECs [6], and we found its transcripts significantly enriched in cluster 0 (Figure 5B,C). We also noted that *Cytl1* was robustly expressed in clusters 3, 4, and 5 as well (Figure 5A–C). Similar to the mouse Cytl1^high^ chondrocytes, the human Cytl1^high^ cluster (cluster 0) showed enrichment for *Chad*, *Wif1*, and *Phlda1* (Figure 5D, Appendix A). Cluster 1 showed enrichment for *Tnfaip6*, which was described as a preHTC marker by Ji et al. [6]. This human Tnfaip6^high^ cluster shared markers (*Tnfaip6*, *Abi3bp*, *Prg4*, *S100a4*, etc.) with the mouse Tnfaip6^high^ and S100a4^high^ clusters (Figure 5A,D, Appendix A) but did not show enrichment for pre-hypertrophic chondrocyte markers including *Ihh*, *Mef2c*, and *Pth1r* [38,39,40] (Figure 5A,D, Appendix A). Chil1 (CHI3L1), which marked RegCs [6], was enriched in cluster 2 (Figure 5A,D, Appendix A). Other markers of murine Chil1^high^ chondrocytes including *Efemp1*, *Prnp*, and *Thbs1* also showed enrichment in this cluster (Figure 5D, Appendix A). Ji et al. identified *Jun* as a marker of HomCs which also showed enrichment for *Fos* and *Fosb* (Figure 5B,D, Appendix A). In mice, these transcription factors were robustly expressed in almost all chondrocyte subtypes, whereas in humans, the expression was restricted to cluster 3 (Figure 5D,E). *Krt16*, a marker of proCs, was primarily expressed by cluster 4, which also showed enrichment for *Il11*, *Ngf*, *Matn3*, and *Smox* like the mouse Krt16^high^ cluster (Figure 5D, Appendix A). Cluster 5 highly expressed the HTC marker *Col10a1* along with *Bhlhe41* and *Wwp2*, while cluster 6 showed enrichment for markers of FCs including *S100a4*, *Col1a1*, *Col1a2*, and *Thy1* (Figure 5D) [6]. FCs also showed enrichment for *Col10a1* along with *Alpl*, *Mef2c*, and *Pth1r*, regulators of chondrocyte hypertrophy as well as biomineralization. Ji et al. also described a small population of cartilage progenitor cells (CPCs) expressing cell-cycle genes like the mouse divCs [6].

This analysis showed that Cytl1^high^, Chil1^high^, Krt16^high^, Tnfaip6^high^, and S100a4^high^ chondrocyte subtypes and divCs exist in both human and mouse joints. Although Ucma^high^ cluster was a major chondrocyte subtype in mice, we could not find an Ucma^high^ cluster in human OA cartilage (Figure 5B), suggesting that this population may be diminished in advanced stages of the disease.

### 3.6. Identification of Injury-Induced Early Molecular Changes in the Articular Chondrocytes

Joint trauma is a major contributing factor to OA. To understand injury-induced early molecular changes in the articular chondrocytes that may contribute to cartilage degeneration, we compared chondrocytes purified from uninjured joints to those isolated from 3-day (3DPI) and 7-day post-injury joints (7DP1). Knee injury was induced by a tibial compression injury model in which the mice develop severe post-traumatic osteoarthritis (PTOA) by 6 weeks post-injury (Figure 6A) [12,13,19,45,46,47]. Histologically, the cartilage appeared normal at 3DPI, but slight proteoglycan loss was observed at 7DPI, as indicated by a lighter Safranin-O staining intensity (Figure 6A). However, scRNA-seq analysis revealed several significant injury-induced transcriptional changes in chondrocyte subpopulations. Like healthy joints, nine chondrocyte subpopulations were identified at 3DPI and 7DPI (Figure 6B–E). These clusters were highly consistent between groups and expressed similar markers in both injured and uninjured joints (Figure 6E and Appendix A). However, we obtained significantly lower numbers of chondrocytes from 3DPI joints compared to other groups, but the numbers of fibroblast and other connective tissue-forming cells were comparable to uninjured controls (Appendix A). Transcripts encoding for enzymes from oxidative phosphorylation pathway (OXPHOS) were significantly enriched in cells from 3DPI (Appendix A). It is likely that the reactive oxygen species (ROS) produced by OXPHOS could have contributed to cellular stress and reduced chondrocyte viability at 3DPI. Most of the chondrocytes we obtained from 3DPI belonged to Tnfaip6^high^ and S100a4^high^ clusters which expressed fibroblast/fibrosis markers including *Col3a1*, *Col1a1*, and *Abi3bp* (Figure 6F and Appendix A).

Differential expression analysis was primarily focused on uninjured controls and chondrocytes from 7DPI as these timepoints had comparable numbers of chondrocytes present in each cluster (Figure 6F). By comparing all chondrocytes (except divCs) from uninjured and 7DPI joints, we identified 51 genes differentially expressed in response to injury. *Col9a1-a3*, *Sox9*, *Prelp*, *Il17b*, and *Cilp* were significantly downregulated, whereas *Mmp3*, *Mmp13*, *Slpi*, and *Col10a1* were significantly upregulated in response to injury (Appendix A). By analyzing cells from each cluster separately, we also determined that *Mmp3*, *Inhba*, *Sfn*, *Il11*, *Ptgs2*, *Dusp2*, and *Mmp13* were upregulated, whereas *Cytl1*, *Il17b*, *Fgfr2*, *Ptch1*, *Dbp*, and *Rrad* were downregulated in various chondrocyte subtypes in response to injury (Figure 7A, Appendix A, Appendix A). Protein-level validation using IHC further confirmed that Mmp3 and Inhba were upregulated, while Cytl1 was down-regulated at as early as 3DPI (Figure 7B). We also observed an expansion in Chil1 expression after injury (Appendix A), suggesting that injury activated Chil1 expression in chondrocytes that do not normally express this gene. Future experimental validation will be required to determine whether some of these markers transcriptionally affected by injury are direct contributors to OA pathogenesis or whether they represent biomarkers of disease progression.

## 4. Discussion

In recent years, considerable efforts have been invested in understanding joint biology and pathophysiology of OA, using joint tissues from both human patients and animal models. However, because most of these studies investigated gene/protein expression at the bulk level of the joint tissue, the contribution of the individual cell type/subtype to joint degeneration was not determined. Recent advances in single-cell sequencing technologies now permit us to profile the transcriptomes of individual cells at an unprecedented resolution, expanding our understanding of molecular responses in health and disease states. In a recent scRNA-seq study, Ji et al. defined seven populations of articular chondrocytes in the human OA cartilage and highlighted some OA progression-associated changes [6]. Chou et al. also examined human specimens using scRNA-seq to investigate molecular crosstalk between cartilage and synovium in OA [7]. Our study extends the findings obtained from individual OA patients by successfully characterizing the cellular and transcriptional heterogeneity of chondrocytes purified from healthy and injured joints, and provides a detailed account of cross-species comparison at single cell level.

Here we give a first account of nine chondrocyte subtypes identified in the articular cartilage of the healthy mouse knee joint and provide detailed molecular definitions of Ucma^high^, Cytl1^high^, Chil1^high^, Mef2c^high^, Krt16^high^, Tnfaip6^high^, S100a4^high^, Neat1^high^, and divCs chondrocyte clusters. All these clusters had distinct transcriptome profiles, including enrichment for specific cartilage ECM proteins and ECM-modifying enzymes (Figure 4E,F, and Figure 5A). We found that Cytl1^high^, Chil1^high^, Tnfaip6^high^, and S100a4^high^ clusters express several genes with signaling and regulatory functions. Ucma^high^ and Krt16^high^ clusters showed enrichment for genes involved in protein synthesis and mRNA metabolism, including numerous ribosomal proteins indicating active protein synthesis. The Ucma^high^ and Krt16^high^ clusters also had a distinct EMC profile with increased expression of several key ECM proteins including *Col2a1*, *Col9a1*-*a3*, and *Acan* (Appendix A). Although the Mef2c^high^ cluster shared several markers with Ucma^high^ and Krt16^high^ clusters, it showed significant enrichment for markers of pre-hypertrophic/hypertrophic chondrocytes including *Ihh*, *Alpl*, *Mef2c*, *Pth1r*, and *Col10a1* [38,39] suggesting that this cluster may represent pre-hypertrophic/hypertrophic chondrocytes (Figure 2C and Appendix A). We could not obtain significant information about the Neat1^high^ cluster, as this cluster had minimal genes enriched in its cells.

We also identified differences in chondrocyte transcriptome relative to other connective tissue-forming cells in the joint including fibroblast, osteoblast, SIF, and endothelial cells. These differentially enriched genes include transcription factors *Nfatc2*, *Runx3*, *Bhlhe40*, *Bhlhe41*, *Lef1*, *Tsc22d1*, *Sox9*, and *Sox5*, which may play a crucial role in chondrocyte development and homeostasis. *Sox5* along with *Sox9* has been shown to regulate chondrogenic pathway [48]. *Sox6*, another known regulator of chondrocyte differentiation was also highly expressed in chondrocytes [48] (Appendix A). Deletion of *Nfatc2* in mice has been shown to cause early onset, aggressive OA affecting multiple joints, suggesting that Nfatc2 plays a key role in maintaining cartilage homeostasis [49]. Runx3 plays an important role in chondrocyte maturation [50]. Lef1 has been shown to regulate *Ihh* expression in pre-hypertrophic chondrocytes and matrix metalloproteinase 13 (*MMP13*) gene expression in OA chondrocytes [51,52]. Further studies are required to understand if *Bhlhe40*, *Bhlhe41*, or *Tsc22d1* play a significant role in maintaining cartilage homeostasis. We also identified several potential OA targets enriched in chondrocytes compared to other cell types including *Scrg1*, *Smpd3*, *Ppa1*, *Tsc22d1*, and *Msmo1*, suggesting that these genes play a role in maintaining articular cartilage integrity (Figure 2A).

Our study also revealed both similarities and differences between mouse and human chondrocyte transcriptomes. We found mouse Cytl1^high^, Chil1^high^, Krt16^high^, S100a4^high^, and divCs clusters to correspond to human ECs, RegCs, proCs, FCs, and CPCs clusters, respectively [6], but gene expression profiles did not show a direct one-to-one correspondence (Appendix A, Figure 5D). However, we identified several markers shared between these human and mouse chondrocyte subtypes. The Tnfaip6^high^ human chondrocytes we identified through the reanalysis of the data of Ji et al. were named ‘pre-HTCs’ in their manuscript, but this cluster did not show enrichment for some known pre-hypertrophic chondrocyte markers such as *Ihh*, *Mef2c*, and *Pth1r*. In our mouse data, pre-hypertrophic markers were expressed predominantly in the Mef2c^high^ cluster. However, the human Tnfaip6^high^ cluster did not share any markers with the mouse Mef2c^high^ cluster. Instead, it shared markers with both the mouse Tnfaip6^high^ and S100a4^high^ clusters. This raises the question whether human Tnfaip6^high^ cluster represented a pre-fibrotic subtype of a pre-hypertrophic phenotype. We also noted the absence of the mouse Ucma^high^ population in the human scRNA-seq OA data. This would suggest that this cluster is either rodent-specific or is lost during OA development. Ucma has been shown to play a cartilage protective role in the context of inflammatory arthritis [53]. A continued survey of mouse joints at later stages of the disease will help determine whether this Ucma^high^ mouse population also disappears as a function of disease progression. Furthermore, several homeostatic genes (*Jun*, *Fos*, *Fob*, etc.) exhibited broad expression patterns across many chondrocyte subtypes in the mouse (Figure 5E), while their expression was restricted to a subset of chondrocytes in human data (Figure 5B,D), suggesting that these differences reflect OA-induced changes in the human articular cartilage.

Wnt signaling is a major regulator of chondrogenic differentiation. Inhibition of Wnt signaling has been shown to enhance chondrogenesis [54], and our group has previously shown that upregulation of Wnt-inhibitor sclerostin in the articular cartilage blunts the outcomes of trauma-induced OA [19]. It has also been shown that human iPSC-derived chondrogenic pellets treated with individual Wnts exhibited increased Col10a1 staining, suggesting that Wnts promote chondrocyte hypertrophy [54]. Consistent with these prior reports, the Mef2c^high^ cluster, which expresses several genes associated with chondrocyte hypertrophy, also expressed high levels of *Wnt5b* (Appendix A). Interestingly, Chil1^high^ cluster showed significant enrichment for several Wnt inhibitors including *Wif1*, *Sfrp5*, *Notum*, *Frzb*, and *Dact1*. The Chil1^high^ cluster (RegCs) showed enrichment for the Wnt signaling pathway in the human chondrocyte data as well [6], suggesting that these cells may mitigate the cartilage-catabolic effects of Wnt signaling through the secretion of several Wnt antagonists.

TGF-beta/Bmp signaling pathway, another major regulator of skeletal development, was enriched in the Cytl1^high^, Tnfaip6^high^, and S100a4^high^ clusters. These clusters showed significant enrichment for *Bmp2* (Appendix A), which is a crucial regulator of chondrogenesis and a potential mediator of cartilage repair [55,56]. The Tnfaip6^high^ and S100a4^high^ clusters also showed significant enrichment for regulators of mesenchymal cell proliferation including *Prrx1*, *Nfib*, and *Msx1* (Appendix A), suggesting that these cells may possess stem/progenitor cell-like properties.

We also identified several injury-induced early molecular changes in chondrocytes, including up-regulation of *Mmp3*, *Mmp13*, *Ptgs2*, *Inhba*, *Sfn*, and *Il11* and down-regulation of *Cytl1*, *Errfi1*, and *Il17b*. Previous studies have identified elevated expression of *Mmp3*, *Mmp13*, *Ptgs2*, *Il11*, and *Inhba* in OA [57,58,59], and our results confirm that articular chondrocytes are a main source of this elevated expression in the injured joint. Furthermore, polymorphisms in matrix metalloproteinase 3 (MMP3), Prostaglandin-endoperoxide synthase 2 (PTGS2), and interleukin (Il)11 have been identified as risk factors for OA [60,61,62], and celecoxib, a selective PTGS2 (COX-2) inhibitor, has been shown to have chondroprotective effects [63,64], strengthening the functional role of these proteins in OA. Sulforaphane (SFN), which was also induced after injury, may have a chondroprotective effect [65]. *Il17b*, a member of the interleukin (IL)17 superfamily of cytokines [66], was downregulated after injury and very little is known about its function in chondrocytes. The dynamic transcriptional changes that we describe in individual chondrocyte subpopulations from the injured joint highlight the importance of elucidating which changes are part of the normal healing process and which ones are disease-promoting in order to effectively develop early therapeutic interventions that will promote healing and prevent cartilage damage after injury.

The joint is a complex structure composed of articular cartilage, subchondral bone, synovium, synovial fluid, joint capsule, menisci, infrapatellar and suprapatellar fat pads, nerves, vasculature, and tensile tissues including tendons and ligaments. It also includes resident immune cells that become activated during injury and tissue repair [67]. Traditionally, OA was perceived primarily as a disease of the articular cartilage; however, emerging evidence suggests that changes in almost all tissues of the joint contribute to OA pathology [68,69]. While our current study focused exclusively on examining transcriptional changes in chondrocyte subpopulations, we recognize that future studies will need to include other cells in the joint microenvironment such as synovial cells, osteoblasts, endothelial cells, mesenchymal cells, and infiltrating immune cells to identify the role played by each cell type in maintenance of joint homeostasis and OA pathogenesis. Although our study highlighted several potential OA targets enriched in these connective tissue-forming cells, an in-depth analysis these cell types as well as immune cells in the joint is necessary to obtain a comprehensive understanding of early molecular changes in the joint that contribute to OA.

While we were able to identify nine subpopulations of chondrocytes, the possibility still exists that more subpopulations are present in the healthy articular cartilage of rodents, as well as in humans. As tissue digests protocols improve, and viable cells are released from densely packed ECM and mineralized tissues like bones and cartilage, we will be able to sequence an exhausting number of cells, expanding our knowledge of rare and novel cell populations. The detailed transcriptional profiling of chondrocytes from injured and uninjured mouse joints allowed us to report here, for the first time, discriminative molecular markers for various chondrocyte subtypes in healthy joints as well as significant transcriptional changes at the single chondrocyte level, as a function of joint injury. These results provide insights into the potential roles some of these genes play in maintaining cartilage homeostasis. Furthermore, the genes we identified as differentially expressed in chondrocytes post injury may play a role in early OA development. Our findings expand our knowledge of cartilage biology and open new avenues for developing improved diagnostic and preventive strategies for mitigating the long-term damaging effects of joint trauma on the articular cartilage.

## Figures and Tables

**Figure 1 cells-10-01462-f001:**
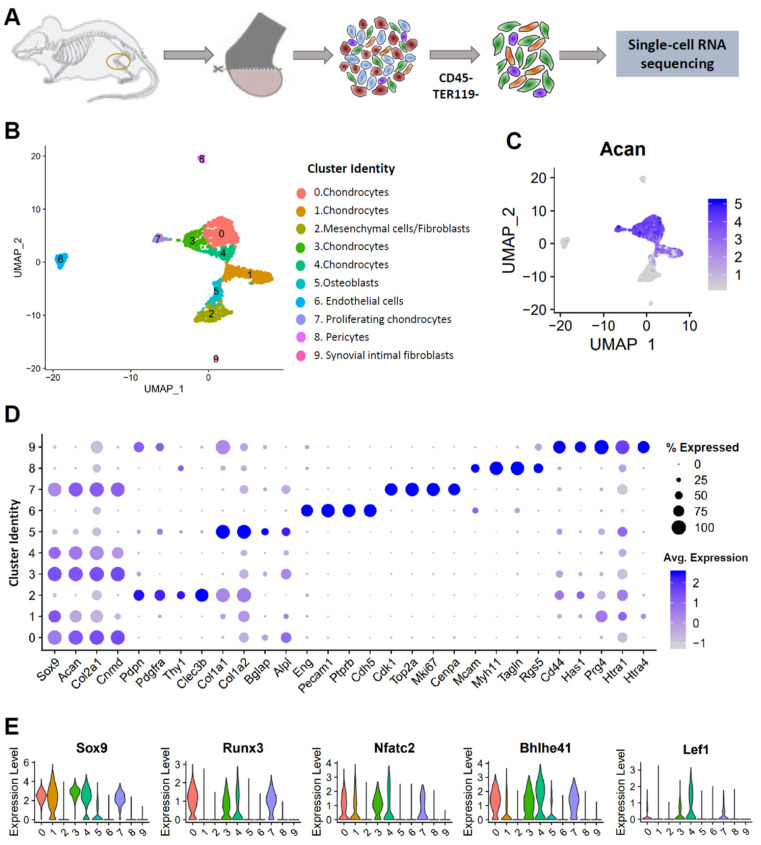
Single-cell analysis of 10-week-old BL6 mouse knee joints. (**A**) Graphical representation of the experimental workflow. Cartilage from mouse knee joints was dissected, dissociated into single cells, and subjected to immune and blood cell depletion. Viable cells from the remaining fraction were sequenced. (**B**) Cell clusters from scRNA-seq analysis visualized by Uniform Manifold Approximation and Projection (UMAP). Colors indicate clusters of various cell types. (**C**) Feature plot showing the expression of chondrocyte marker *Acan*. (**D**) Dot plot showing the expression of selected markers of various cell types. Dot size represents the % of cells expressing a specific marker, while the intensity of color indicates the average expression level for that gene, in that cluster. (**E**) Violin plot showing the expression of key transcription factors enriched in chondrocytes.

**Figure 2 cells-10-01462-f002:**
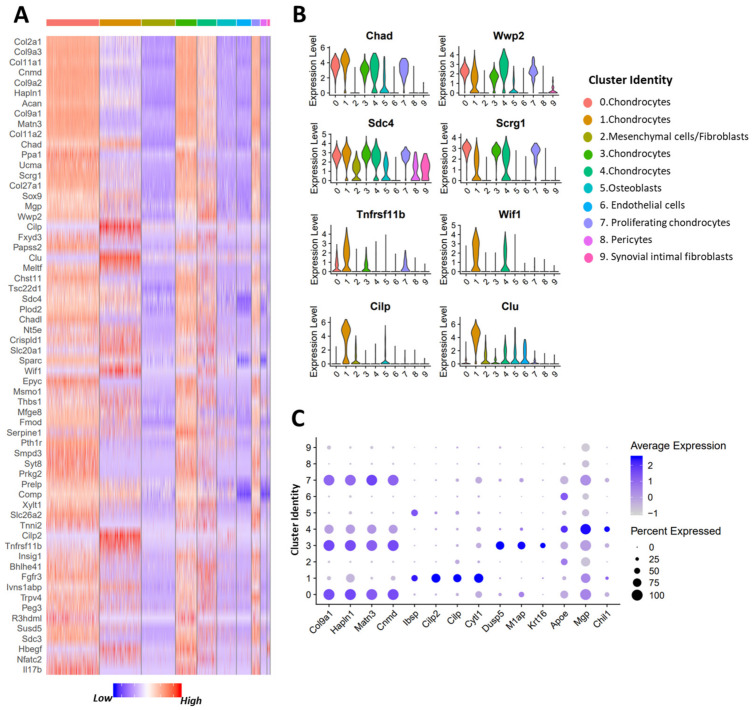
Identification of potential OA targets enriched in chondrocytes. (**A**) Heatmap showing potential OA targets enriched in chondrocyte clusters compared to other connective-tissue forming cell types in the joint. (**B**) Violin plot showing the expression of selected OA targets that are expressed in all chondrocyte subtypes and OA targets with a restricted expression pattern. (**C**) Dot plot showing the expression of selected markers of various chondrocyte clusters. Dot size represents the fraction of cells expressing a specific marker and color intensity indicates the average expression level in that cluster.

**Figure 3 cells-10-01462-f003:**
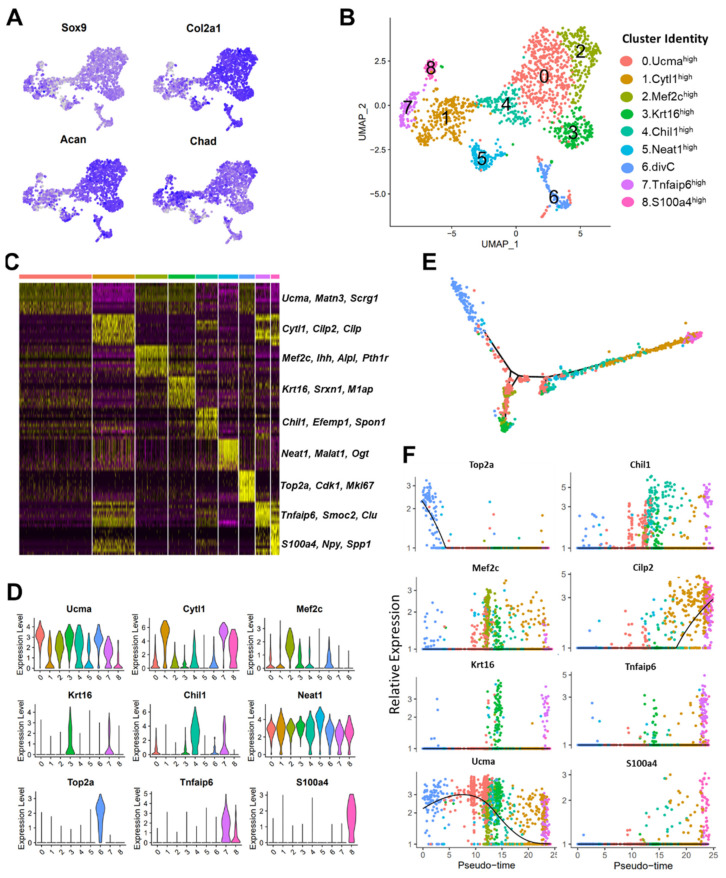
Characterization of chondrocyte subtypes. (**A**) Feature plots showing the expression of key chondrocyte markers. Blue: high expression, grey: low expression. (**B**) UMAP plots of various chondrocyte subtypes in mouse knee joints. Colors indicate clusters of various cell types with distinct gene expression profiles. (**C**) Heatmap showing the scaled expression of top genes differentially expressed in each cluster. (**D**) Violin plots showing the expression of selected markers of various chondrocyte subtypes. (**E**) Monocle pseudotime trajectory colored based on chondrocyte clusters in (**A**). (**F**) Expression of chondrocyte subtype markers on a pseudotime scale (colored based on clusters in (**A**)).

**Figure 4 cells-10-01462-f004:**
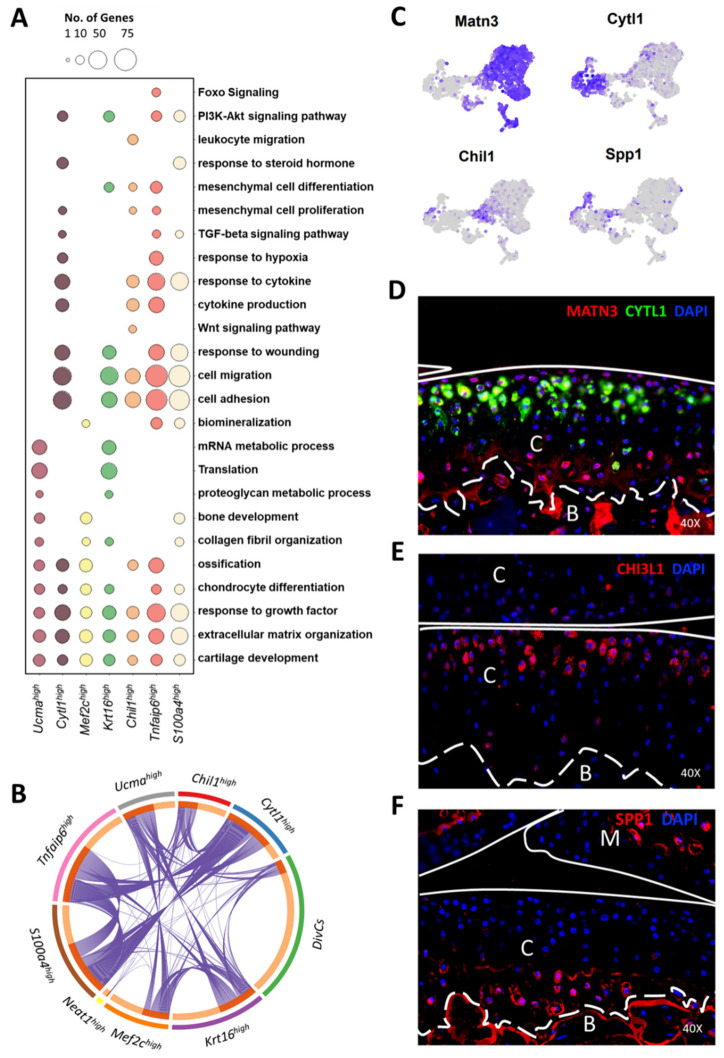
Functional characterization of chondrocyte subtypes. (**A**) Enriched ontologies for each chondrocyte subtype. Circle size indicates the number of genes associated with each category. (**B**) Circos plot showing overlap between genes enriched (>1.25-fold) in each cluster compared to all other clusters. The purple curves link identical genes. (**C**) Feature plots showing the expression of *Matn3*, *Cytl1*, *Spp1*, and *Chil1*. (**D**) Protein-level expression of Matn3 (red) and Cytl1 (green). (**E**) Protein-level expression of Chil1/CHI3L1 (red). (**F**) Protein-level expression of Spp1 (red). DAPI (blue); C: cartilage; B: bone; M: meniscus.

**Figure 5 cells-10-01462-f005:**
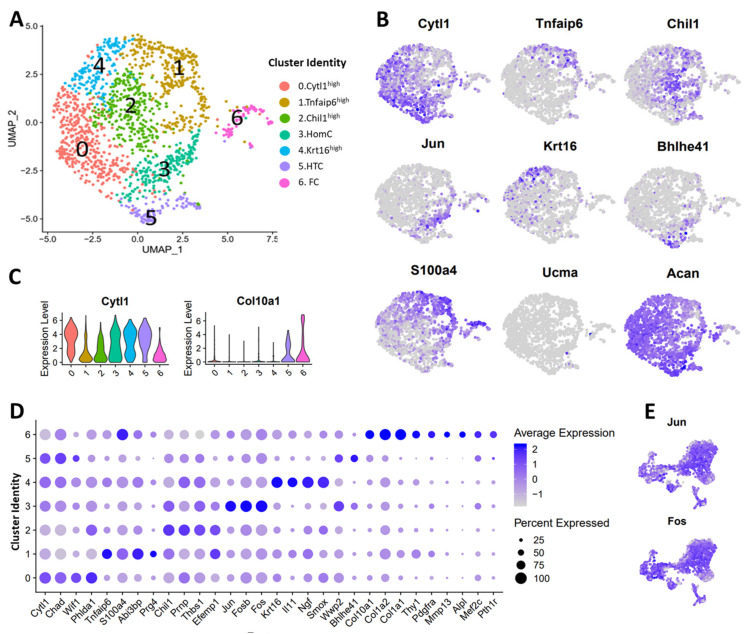
Comparison of human and mouse articular chondrocytes. (**A**) UMAP plots of various chondrocyte subtypes in human osteoarthritic knee joints. Colors indicate clusters of various cell types with distinct gene expression profiles. (**B**) Feature plots showing the expression of key chondrocyte markers in human chondrocytes. Blue: high expression, grey: low expression. (**C**) Violin plot showing the expression of *Cytl1* and *Col10a1* in human chondrocyte subtypes. (**D**) Dot plot showing the expression of selected markers of various clusters. Dot size represents the fraction of cells expressing a specific marker in a particular cluster and intensity of color indicates the average expression level in that cluster. (**E**) Feature plots showing the expression *Jun* and *Fos* in mouse chondrocytes.

**Figure 6 cells-10-01462-f006:**
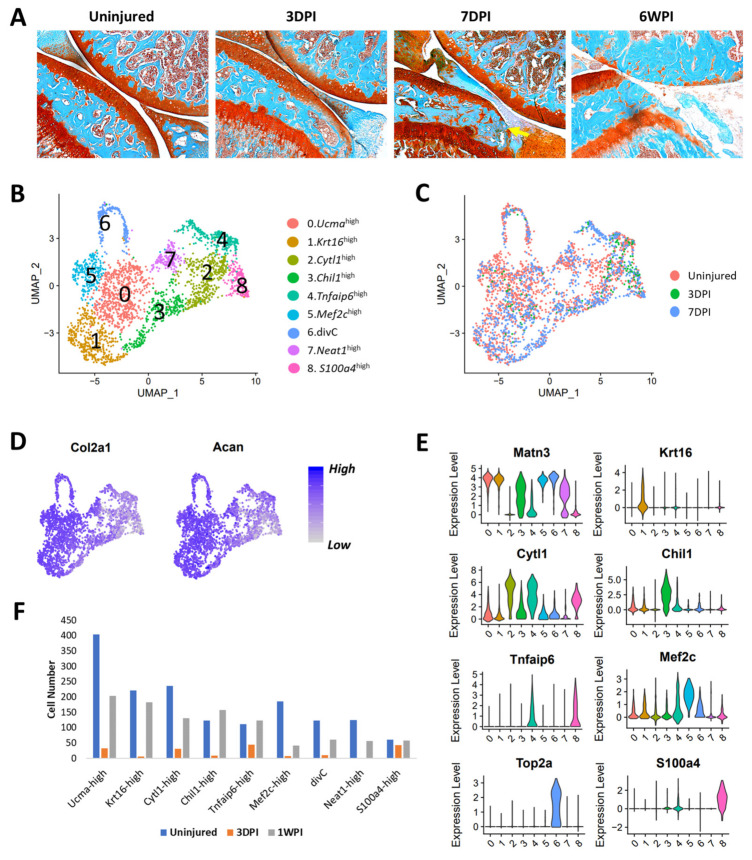
Injury-induced changes in chondrocytes. (**A**) Histological assessment of uninjured and injured joints at 3 days (3DPI), 7 days (7DPI), and 6 weeks (6WPI) post-injury using Safranin-O and Fast Green staining. Severe cartilage degeneration was observed at 6WPI and minor proteoglycan loss was observed at 1WPI (yellow arrow). (**B**) UMAP plots of various chondrocyte subtypes identified in uninjured, 3DPI, and 7DPI joints. Colors indicate clusters of various cell types with distinct gene expression profiles. (**C**) UMAP plots of various chondrocyte subtypes identified in uninjured, 3DPI, and 7DPI joints. Colors indicate cells from each experimental group. (**D**) Feature plots showing the expression of chondrocyte markers Col2a1 and Acan; high expression (blue), low expression (grey). (**E**) Violin plot showing the expression of chondrocyte subtype markers. (**F**) Sequenced cells per chondrocyte cluster.

**Figure 7 cells-10-01462-f007:**
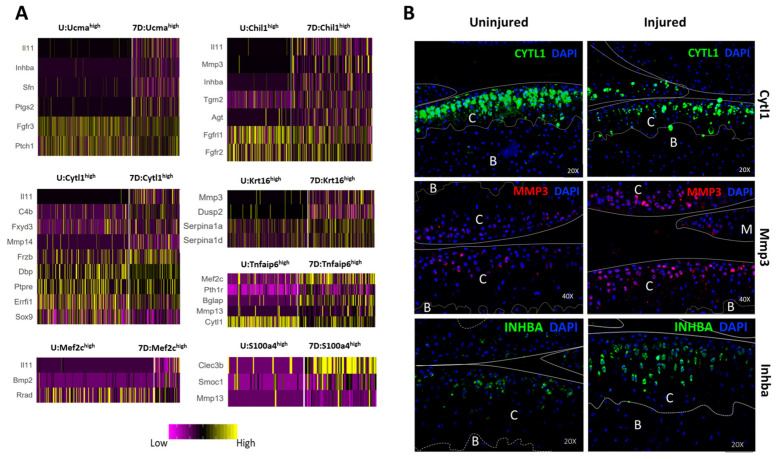
Injury-induced early molecular changes in articular chondrocytes. (**A**) Heatmaps showing key differentially expressed genes in each chondrocyte cluster. U: uninjured; 7D: 7 days post-injury. (**B**) Immunohistochemistry analysis showing protein-level expression of Cytl1 (20×, Mmp3 (40×), and Inhba (20×) in uninjured and 3DPI joints. B: bone; C: cartilage, M: meniscus.

## Data Availability

Single-cell sequence data that support the findings of this study are available through the National Center for Biotechnology Information Gene Expression Omnibus (GSE172500).

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
