# Peer review of "Single-Cell RNA-Seq Reveals Transcriptomic Heterogeneity and Post-Traumatic Osteoarthritis-Associated Early Molecular Changes in Mouse Articular Chondrocytes"

_cells, 2021, doi:10.3390/cells10061462_

Round 1

Reviewer 1 Report

I would think the current version of the manuscript is at the acceptable level.

Author Response

Thank you for your positive feedback

Reviewer 2 Report

I really like the the manuscript, eventhough I do not appreciate single-RNA seq data papers. The manuscript has clear novelty and describes an identification of various chondrocyte subtypes in healthy state and during post-traumatic osteoarthritis. 

I would like to discuss several following points with the authors:

  1. Experimental design: Cell isolation require long incubation times with enzymes. Such procedure may result in changes of transcriptome of isolated cell. Previsuoly, some groups applied various chemical agents that blcok transcription and stabilize mRNA. Could authors discuss how this may affect the outcome of their study?
  2. Taking into account the isolation procedure and failure to isolate enough cells on 3dpi (Fig. 3F), I was wondering how reproducible is the isolation itself. Can authors discuss this and may be show some data? Also here, I did not fully understand the inclusion of 3dpi analysis, since authors failed to isolate sufficient amount of cells. I would consider removal of the data, given the technical issues with cell isolation.
  3. Histological pictures presented on various figures look quite convincing. Would it be possible to include quantification and respective statistical analysis and also to mention the number of animals used for the analysis? 
  4. Authors also provide anaylsis of human scRNA-Seq data (Fig. 5). However, these data are not mentioned neither in title, nor in the abstract. Authors may consider it, as it may increase visibility of their work.

Author Response

we thank the reviewer for the insightful comments and suggestions, the point by point response is included in the attached file

Reviewer 3 Report

In the manuscript titled “Single-cell RNA-seq reveals transcriptomic heterogeneity and post-traumatic osteoarthritis-associated early molecular changes in mouse articular chondrocytes”, the authors tried to search the molecular regulated in early stage of osteoarthritis by single-cell RNA-seq.

  1. The abbreviation “ACL” should be explained as "anterior cruciate ligament (ACL)" in line 98 of page 3, where "ACL" appeared for the first time in the manuscript.

  1. It is often difficult to understand what kind of genes are considered in Figure 2 and the following figures. For all the genes considered in this study, you should explain either their full names or the cell types for which the genes are used as markers. Above all, it may be easily understood if “Spp1” (P.10, line 345) is replaced by “osteopontin (OPN) / secreted phosphoprotein 1(Spp1)”.

  1. In Figures 3B and 6B, the font sizes of the numbers should be as large as those in Figure 5B.

  1. The following sentence in lines 362-367 of page10 should be shortened: “This analysis suggested that Tnfaip6high, S100a4high and Cytl1high clusters are closer in developmental time while Krt16high and Mef2chigh clusters are developmentally closer to Ucmahigh cluster which begged the question whether the physical relationship to each other was topologically similar in the articular cartilage contributing to distinct functions of chondrocyte subpopulations based on their physical locations within the joint.”

  1. In Figures 4D-F and 7B, it would be better to place the texts outside the pictures, otherwise the font size should be larger.

  1. In Figures 4F, why did you perform immunohistochemistry for SPP1, not for S1004a; while in other figures, Figs. 4D and 4E, it was performed for the proteins which labeled the corresponding clusters?

  1. In Figures 4D-F and 7B, the negative controls without primary antibodies should be indicated.

  1. The abbreviation “Wif1” should be explained as “Wnt inhibitory factor 1 (Wif1)" in line 298 of page 7 or line 304 of page 8, while in lines 407-408 of page 12, the abbreviation "Wif1" may be used.

  1. The abbreviation “PTOA” should be explained (p. 14, line 500).

  1. 10. In Figure 6A, you should explain what the yellow arrow indicates.

  1. It would be clearer if the color of the cluster and the order of the list in Figures 6B coincide with those in Figures 3B.

  1. Should "injured and 7DPI joints" (p. 15, lines 523-524) be "uninjured and 7DPI joints”?

  1. It would be better if the immunohistochemistry is performed for the proteins which are encoded by the genes used for labeling the clusters, as for CYTL1 and CHIL1 in Figures 7B and S10.

  1. Should "jury" (p. 16, lines 530) be "injury”?

  1. You stated that Mmp3 and Inhba were upregulated in various chondrocyte subtypes in response to injury. However, are there no possibilities that the expression level of Mmp3 and Inhba are high in some clusters in particular? I suggest that you perform the double immunofluorescent staining for Mmp3 or Inhba and for the proteins that are encoded by the genes used for labeling clusters.

I hope these comments will be helpful.

Author Response

we thank the reviewer for the insightful comments and suggestions, a point by point response to reviewers is in the attached file
